# Healthcare professionals' perceptions about implementing accreditation as a strategy to improve healthcare quality and organisational performance: a cross-sectional survey study

**Mohammad J. Alhawajreh**[1]*, **William J. Jackson**[2], **Audrey S. Paterson**[2]

**1** Health Services Management, Middel East University, Amman, Jordan, **2** Business School, University of Aberdeen, Aberdeen, Scotland, United Kingdom

* m.hawajreh@meu.edu.jo; alhawajreh@yahoo.com

## Abstract

### Objective

This study aims to investigate the healthcare professionals' perceptions about the impact of accreditation as a strategy to improve the quality of healthcare services and organisational performance in public hospitals in Jordan.

### Design

A cross-sectional survey.

### Setting

Four accredited public hospitals located in three different geographical regions in Jordan.

### Participants

A total of 500 healthcare professionals, including both clinical and non-clinical staff, who worked at the selected hospitals. The data analysis included valid responses from 74.4% of the participants.

### Methods

A web-based questionnaire was applied. To investigate the relationship between the quality results scale (dependent variable) and the other survey scales (independent variables), multiple regression analysis was performed.

### Results

The study showed that the impact of accreditation on the quality of healthcare services was almost high across all scales; the Quality Results and Accreditation Impact scales received the highest mean scores, suggesting that healthcare professionals have a

**Data availability statement:** All relevant data are within the paper and its Supporting information file.

**Funding:** The author(s) received no specific funding for this work.

**Competing interests:** The authors have declared that no competing interests exist.

positive perception towards accreditation. Healthcare professionals agree that accreditation has improved the quality of healthcare services and hospitals' performance in a wide range of clinical and administrative aspects. There was a significant positive correlation between healthcare professionals' perception of Quality Results (P < 0.001) (dependent variable) and their perception of all other accreditation survey scales (independent variables). Regression analysis showed a significant relationship between Quality Results and Customer or Patient Satisfaction (P < 0.001), Management and Leadership (P = 0.030), and Accreditation Impact (P = 0.001) scales, and these were the most important accreditation predictors of, better quality results.

## Conclusion

The majority of healthcare professionals view accreditation positively, seeing its potential to improve the quality of healthcare and organizational performance. However, there are nuanced variations between administrative and clinical professionals about the aims and objectives of using accreditation as a strategy for healthcare quality improvement. Future research is needed to explore the long-term impact of accreditation and the contextual factors that impact the perceptions of healthcare professionals on accreditation, particularly among different healthcare professional groups. This will help provide policy direction and improve understanding of accreditation's impact.

## Introduction

Worldwide, the healthcare industry faces unprecedented challenges from ageing populations, resource constraints, rising public awareness, and more recently, the challenges of a global pandemic [1,2]. These issues place a significant burden on healthcare systems while also reducing their quality and efficiency. To address these challenges, huge investments and persistent efforts have been undertaken to improve the quality and safety of healthcare provided globally [3–5]. More than two decades ago, in its ground-breaking report "To Err Is Human," the Institute of Medicine (IOM) (2000) concluded that the majority of medical errors are caused by flawed health systems and processes rather than by individuals. Consequently, the need for process improvement and safety initiatives in healthcare has addressed the potential use of various quality improvement strategies [6,7].

Accreditation is one of the most widely used quality strategies for improving health systems and healthcare outcomes [8–10]. Accreditation is defined as: *"a public recognition by a healthcare accreditation body of the achievement of accreditation standards by a healthcare organisation, demonstrated through an independent external peer assessment of that organisation's level of performance in relation to the standards."* [11]. Accreditation has become a critical component of healthcare systems in over 70 countries around the world that are members of the International Society for Quality in Health Care (ISQua) [12,13]. Accordingly, accreditation and quality improvement initiatives have captivated the interest of researchers and are increasingly playing a part in debates and daily activities within the healthcare sector worldwide [6,10,14].

While healthcare accreditation programs are voluntary in many countries, for example, the US, Germany, and Australia; other countries, such as France, Italy, and Scotland, have made participation in accreditation legally mandatory for healthcare organizations [11,17]. In developed countries like the US, Canada, and Australia, healthcare accreditation programs are often well-structured and rigorously enforced. These countries have advanced accreditation

bodies that enforce specified evidence-based standards, conduct periodic reviews, and have a transparent reporting system. This generally results in measurable improvements in patient outcomes, decreased number of medical errors, and higher overall healthcare quality [9]. On the other hand, the landscape of accreditation in developing countries like Jordan, Saudi Arabia, and Lebanon presents a more nuanced picture. While these countries have made advancements in implementing accreditation programs, challenges such as limited resources, varying levels of expertise, cultural differences, and political instability with a high number of refugees could inhibit their effectiveness [9,20].

In the literature, several studies show that health care organisations (HCOs) participating in accreditation programmes outperform those not in terms of strengthening the healthcare system, improving clinical healthcare, and optimising financial performance [15–20]. However, other work on the value and merits of accreditation has yielded mixed results with scant evidence supporting its actual effect on quality of healthcare services and clinical outcomes [21–25]. Accreditation has also had unintended consequences, such as increased financial expenses, staff time demands, and diverting resources away from clinical care [26–30].

Overall, the literature on the evidence of accreditation's effectiveness in enhancing quality remains inconclusive, with contradicting findings on its effect on the performance of HCOs [20,31,32]. The literature also shows limited research on accreditation in developing countries, particularly in the Eastern Mediterranean Region (EMR), compared to developed countries such as the United States, Canada, and Australia [33,34]. These inconsistent results about the impact of accreditation on the quality of healthcare services and the lack of research on healthcare accreditation in developing countries, highlight the pressing need for more rigorous research in this area, prompting several calls to investigate whether it truly improves the quality of healthcare services as well as to strengthen the evidence base.

To our knowledge, this is the first empirical study in Jordan's public hospitals and aims to fill important gaps in the current literature by investigating healthcare professionals' perceptions and views on accreditation implementation and its impact on improving healthcare quality and organisational performance in public hospitals.

## Background

Since the mid-1990s, accreditation programs have rapidly multiplied in developed and developing countries, with North America and Europe being the most affected. The World Health Organization (WHO) reports that the US, Canada, the UK, Australia, and France accreditation programs have had the most influence on global healthcare accreditation standards [1,33].

In 2000, the WHO conducted a global survey to verify the status of healthcare accreditation programs in the Eastern Mediterranean Region (EMR), finding no existing programs. Consequently, several initiatives for healthcare quality improvement were taken into account in national health policies and quality plans [15]. Recently, the WHO/Eastern Mediterranean Regional Office (EMRO) conducted a regional survey to follow the evolution of the quality improvement initiatives in the EMR and explore the embedding of quality improvement initiatives in the EMR and found that accreditation is increasingly being employed as a strategy for government regulation to ensure the quality and safety of delivered healthcare [33].

However, a study by Braithwaite et al. (2012), comparing the nature of healthcare accreditation programs between the higher-income countries (HICs) and low- and middle-income countries (LMICs), found that around 60% of accreditation programs in LMICs are regulated and managed by the government, compared to only 8% of participants in HICs. This indicates a national response to insufficient resources and an effort to ensure the sustainability. Accreditation organizations play a key role in evolving the government's role in providing a quality and safety agenda [60].

Jordan, like other EMR countries such as Lebanon and Saudi Arabia, is implementing accreditation as a strategy to improve healthcare quality and safety and strengthen healthcare system performance. In 2007, the Jordanian Ministry of Health (MOH) established the Health Care Accreditation Council (HCAC) in collaboration with international partners (e.g., the United States Agency for International Development (USAID)). The HCAC has gained international recognition for its role in improving healthcare services and promoting patient safety. It became the first entity in the EMR to be accredited by the International Society for Quality in Healthcare (ISQua) [35]. Since 2008, numerous HCOs have been accredited by the HCAC [36].

Accreditation is a costly healthcare project, with various financial expenses typically associated with its implementation process [37,38]. In Jordan, a low- to middle-income country, health facilities participating in the accreditation, particularly those belonging to the MOH, receive most of their funding from grants and external sources like USAID [35]. Therefore, careful evaluations are crucial for efficient resource management. However, there is a lack of evidence of effective resource utilisation to promote quality healthcare through accreditation in Jordan [39,40]. This is partly due to the relatively new nature of the HCAC accreditation programme, with research on its effectiveness still in its early stages. Therefore, it is essential to investigate this issue within the public healthcare sector in Jordan.

## Methods

### Study design and setting

This cross-sectional survey was undertaken in public hospitals in Jordan. A sample of four cases was chosen according to the following criteria: Firstly, geographical and cultural locations, which represent Jordan's three different geographical regions (North, Central, South), and the four largest governorates in the country in terms of high population concentrations, and these are urban areas with easy access. These governorates represent the diversity of Jordan's hospitals at the time of selection. Secondly, public hospitals' organizational size. According to Pomey et al. (2010), large-sized hospitals often encounter challenges in implementing quality improvement interventions, such as accreditation emphasizing the importance of hospital size [17]. Thus, in order to compare hospitals that provided comparable service and healthcare characteristics, hospitals were classified by size as follows: two large-sized hospitals (>200 beds) and two medium-sized hospitals (101-200 beds). Finally, according to Pomey et al., (2010), improvements within healthcare organizations (e.g., hospitals) varied with the number of years the hospitals had spent engaged in accreditation [17]. Therefore, the hospitals were categorized based on their accreditation status by the national accreditation program (HCAC).

Two hospitals (case 1 and case 2) were early accredited in 2011 and successfully reaccredited for four cycles; the HCAC accreditation is valid for two years, while the other two (case 3 and case 4) were newly accredited in 2019 and 2020, respectively [41]. Applying these criteria allows the survey to capture and describe a comprehensive picture of the impact of accreditation on healthcare quality and the factors affecting accreditation in Jordan. The study used multiple cases to gain a comprehensive understanding and maximise lessons learned about the subject, as evidence from multiple cases is often considered more conclusive and strengthens data validity [42]. The main characteristics of the retained case study sites are illustrated in (S1 Appendix). The article is structured considering the SQUIRE guidelines for reporting system-level work (accreditation) to improve the quality, safety, and value of healthcare (S2 Appendix).

### Participants

Considering the relevance of investigating all healthcare professionals' perceptions on the impact of the accreditation process, there were no exclusions based on healthcare job

category or professional groups eligible to participate in the survey. The target population included both clinical staff (e.g., physicians and nurses) and non-clinical staff (e.g., managers and department heads) working at the selected public hospitals. Participants were only eligible if they had worked in the accredited hospitals for at least three years before the accreditation award date. This was based on the maximum period of preparation for the accreditation process being three years from the date of hospital registration with the HCAC accreditation programme [41].

The study used a multistage stratified sampling design to select participants who closely represented the target population and could generalise results to the broader population. A list of targeted staff individuals was prepared with Human Resources (HR) arrangements at participating hospitals. The list was divided into strata based on professional job categories, such as nurses, physicians, pharmacists, and administrators. A simple random sample was employed within each stratum to select subgroups of interest.

The study involved 2002 eligible staff from four participating hospitals. To ensure a 95% confidence level and accurate representation of the target population, the study assumed a 5% confidence interval and 50% degree of variability. Using a predefined table [43], the minimum sample size needed was 333 participants across the four public hospitals. However, due to non-response and non-reachable reasons, the sample size was suggested to be increased by up to 50% [44]. As a result, 500 questionnaires were distributed across the selected hospitals. The sample size required for each participating hospital in the study is presented in (S3 Appendix).

## Measurement tool

A validated web-based questionnaire adapted from previous studies conducted in Lebanon was used to assess healthcare professionals' perceptions and views on the implementation of accreditation for quality improvement in healthcare delivery, developed by El-Jardali and colleagues [9]. However, since Lebanon and Jordan have a comparable cultural and healthcare system, there was no need for questionnaire customization.

The data collection questionnaire focused on five main sections: quality of care, accreditation impact, staff involvement in accreditation, awareness of the accreditation process, and demographic information. The questionnaire included nine scales, totalling 81 items, including Strategic Quality Planning (seven items), Human Resource Utilization (six items), Quality Management (six items), Quality Results (five items), Customer or Patient Satisfaction (seven items), Management and Leadership (nine items), and Accreditation. Accreditation Impact (14 items), Staff Involvement in Accreditation (22 items), and Awareness of the Accreditation Process (five items). The survey used a 5-point Likert scale to assess healthcare professionals' perceptions of hospital accreditation implementation. The questionnaire took approximately 20 minutes to complete and was converted to a web-based format using the 'Snap Survey' platform (http://www.snapsurveys.com). A copy of the questionnaire is presented in (S4 Appendix).

## Ethical approval

Ethical approval was provided by the University of Aberdeen and the Institutional Review Board (IRB) and Research Ethics Committee at the MOH in Jordan (IRB number: MBA-6953/2021). The SNAP online questionnaire included an information sheet and informed consent for participants. They were asked to check a box to confirm their understanding of the information and agree with the formed consent topics. If they didn't want to participate, they could close the browser. The survey was voluntary and anonymous, with no monetary incentives. Participants were all over 16, and no vulnerable adults were at risk. Patients or the public were not involved in the research's design, conduct, reporting, or dissemination plans.

### Recruitment

In January 2022, healthcare professionals were asked to complete the web-based questionnaire. For an efficient data-collecting procedure and a high response rate, participants received a pre-notification email about the study a few days before the survey distribution. Participants were notified via email about the survey, followed by invitations from contact persons in each hospital. The emails included a cover letter, a hyperlink to the questionnaire, instructions for access, and contact information for researchers and hospital points of contact. A reminder email was sent one week after the survey invitation to thank staff who completed it and encourage others. A second reminder was sent two weeks later, and a third reminder was considered one month later to boost responses. The main researcher conducted at least three visits to each hospital, meeting heads of quality improvement departments and management staff to encourage staff to complete the survey. Data collected was downloaded to the researcher's computer and saved in a password-protected share folder, accessible only to the researchers. The data collection period lasted five months, from January 7, 2022, through May 23, 2022.

### Statistical analysis

All statistical analyses were performed using IBM SPSS/v.27 at a significance level of 0.05. Statistical reporting in of this study is in line with the SAMPL guidelines for statistical reporting [45]. The surveys' data were downloaded from Snap Webhost and double-checked for accuracy. Respondent characteristics were described using frequencies and means (±SD), and internal reliability was assessed using Cronbach's alphas test. The study analysed respondents' perspectives on each survey scale by computing scores for each item by calculating items and dividing by the number of items. A score ranged from 1 to 5, with higher scores indicating greater agreement with the scale. No missing data was found in the Web surveys. The survey data was not normally distributed; therefore, the assumption of normality was violated in this study, and nonparametric statistical tests were used to test the dataset. The mean rank of each scale was compared across different demographic categories using a Kruskal-Wallis test to break down survey scale results by demographic variables. When the overall p-value of the Kruskal-Wallis test was statistically significant, then the Mann-Whitney U test was applied to determine which groups are different from others. Spearman Correlation was used to investigate the relationship between the Quality Results scale (dependent variable) and respondents' perceptions of other survey scales (independent variables).

The study used a simple linear regression model to examine if the accreditation survey's eight scales (independent variables) significantly predicted the Quality Results scale (dependent variable) as perceived by healthcare professionals. Multiple linear regression analysis was used to examine the overall effect of the survey scales on quality results (model 1). It was also used to estimate the association between independent variables and dependent variable, accounting for confounding variables like gender, age, working experience, education level, occupation categories, and hospital size, to determine which variable(s) would be considered as better-quality results predictors (model 2).

## Results

The study had a 74.4% response rate, with 372 out of 500 valid responses included in the data analysis. The data file did not contain any incomplete or missing questionnaires, as the web surveys were designed to notify participants of unanswered questions before they proceeded. The majority of respondents (59.1%) were female; (50.3%) were aged between 30 and 45 years; (31.2%) had worked at their hospital for 5 to 10 years. A majority of respondents (67.7%) had

a bachelor's degree, with (38.7%) being nurses, followed by physicians (23.1%) and administrators/managers (16.4%). The demographic characteristics of the study sample are presented in Table 1.

## Distribution of the mean scores and Cronbach's Alpha of the survey scales

The mean scores computed for the survey scales and Cronbach's Alpha are outlined in Table 2. It is clearly seen that the respondents' evaluations of the impact of accreditation on the quality of healthcare services were almost high on all scales. While the Quality Results received the highest mean score (3.70), the Human Resource Utilization had the lowest mean score (3.32). The Cronbach alpha coefficients for scale items ranged between 0.827-0.963, and the questionnaire's overall internal reliability was 0.984 thus indicating high consistency and reliability.

## Participants' views on different scales of the accreditation survey

Full details of the participant's responses to the accreditation survey's various scales and sub-scales are given in (S5 Appendix).

**Table 1. Demographic characteristics of the sample (n = 372).**

|  | N | % |
|---|---|---|
| **Total** | 372 | 100 |
| **Gender** | 152 | 40.9 |
| Male | 220 | 59.1 |
| Female |  |  |
| **Age of respondents** | 118 | 31.7 |
| < 30 years | 187 | 50.3 |
| 30 – 45 years | 41 | 11 |
| 46 – 55 years | 26 | 7 |
| > 55 years |  |  |
| **Working experience in years** | 95 | 25.5 |
| 3 – < 5 | 116 | 31.2 |
| 5 – 10 | 100 | 26.9 |
| >10 – < 15 | 61 | 16.4 |
| > 15 |  |  |
| **Level of education** | 66 | 17.7 |
| Diploma (2 years) | 252 | 67.8 |
| Bachelor's degree | 54 | 14.5 |
| Master's and PhD degree |  |  |
| **Hospital name** | 80 | 21.5 |
| Case1 | 44 | 11.8 |
| Case 2 | 105 | 28.2 |
| Case 3 | 143 | 38.4 |
| Case 4 |  |  |
| **Occupational categories** | 12 | 3.2 |
| Director of the hospital | 144 | 38.7 |
| Nurse | 86 | 23.1 |
| Physician | 19 | 5.1 |
| Pharmacist | 7 | 1.9 |
| Social Worker | 43 | 11.6 |
| Allied health technician | 61 | 16.4 |
| Administration/Management |  |  |
| **Hospital size** | 149 | 40 |
| Medium | 223 | 60 |
| Large |  |  |

Table 2. Results of mean scores and Cronbach's Alpha on survey scales.

| Scale | No. of items | Mean ± (SD) | Cronbach's Alpha |
|---|---|---|---|
| Strategic Quality Planning | 7 | 3.55 ± (0.61) | 0.827 |
| Human Resource Utilization | 6 | 3.32 ± (0.78) | 0.858 |
| Quality Management | 6 | 3.47 ± (0.67) | 0.864 |
| Quality Results | 5 | 3.70 ± (0.47) | 0.910 |
| Customer or Patient Satisfaction | 7 | 3.55 ± (0.62) | 0.877 |
| Management and Leadership | 9 | 3.48 ± (0.67) | 0.927 |
| Accreditation | | | |
| Accreditation Impact | 14 | 3.64 ± (0.60) | 0.955 |
| Staff Involvement in Accreditation | 22 | 3.54 ± (0.62) | 0.963 |
| Awareness of the Accreditation | 5 | 3.56 ± (0.65) | 0.852 |
| **All Scales** | **81** | | **0.984** |

For strategic quality planning, 88.7% of respondents demonstrated that middle managers play a key role in setting quality improvement priorities in their hospital. Nonetheless, only 45.4% of respondents believe employees are involved in developing quality improvement plans. Concerning human resource scale, about 73.4% of respondents agreed that inter-departmental cooperation to improve service quality is supported and encouraged, but only 40.1% received reward and recognition for improving quality. For the quality management scale, despite 81.2% stating their hospital has effective policies in place to support improvement of care and services, only 42.1% believe the hospital designs quality into new services.

Regarding the quality results, despite financial constraints, 82.2% of respondents believe the hospital has consistent and measurable improvements in customer satisfaction, clinical support department services (e.g., laboratory), and patient care quality (e.g., medical and paediatric patients) over the past year. However, 48.2% believe the hospital has made consistent improvements in administration services like finance and HR.

In terms of patient satisfaction, about 89% of respondents believe that employees promptly resolve patient complaints, and over 75% of respondents state that the hospital effectively assesses current patient needs and expectations, while only 37.4% believe the hospital's feedback system effectively assesses future patient needs and expectations. For management and leadership, the majority of respondents (80%) believe senior executives are actively working to improve the quality of care and services, based on feedback from staff, customers, and the accrediting body. However, only 65.6% believe they are the primary driving force behind these efforts. Less than half of respondents believe they allocate sufficient resources or consistently participate in quality improvement activities.

On the scale of accreditation impact, 80% of respondents believe accreditation enhances patient care, staff motivation, teamwork, collaboration, and values among hospital professionals and is a valuable tool to implement hospital changes. However, only 47.8% have been informed of the recommendations made since the last survey, indicating a need for further improvement in hospital implementation. Regarding staff involvement in the accreditation scale, the majority of respondents surveyed had a favourable opinion of the impact of accreditation on their hospital, with over 80% stating it improved multidisciplinary working, healthcare standards, and overall hospital performance. However, only 60% of respondents were encouraged to participate in the accreditation process, and less than half had the opportunity to voice their concerns. Overall, accreditation is considered a worthwhile process by the majority of respondents.

In terms of the awareness of the accreditation scale, over 90% of respondents indicated that hospital staff are aware that the accreditation process is underway, with 70% believing that staff members are aware of the accreditation process's aims and objectives and agreed it's worthwhile. However, only 53.2% of respondents indicated that patients are aware that the process is underway.

## Breakouts of results of the survey scales by demographic variables

The survey showed no significant gender differences across all scales, except for Quality Management and Customer/Patient Satisfaction scales. However, age, work experience, and education level significantly impacted the mean ranks. It's great to know that respondents aged 46-55 had higher mean ranks on all scales, while those with over 15 years of work experience and postgraduate degrees had higher mean ranks. This indicates that demographic factors such as age, work experience, and education level play a significant role in the overall survey results.

The study found no significant differences in hospital size across all scales, suggesting that hospital size has no effect. However, for the occupational categories, the mean rank was significant in the following scales: Human Resource Utilization, Quality Management, Customer Satisfaction, Management and Leadership, Staff Involvement in Accreditation, and Awareness of Accreditation, with the highest scores among hospital directors, administrative and management staff, and nurses, respectively. The associations of each survey scale with various categories of demographic variables are summarised in (S6 Appendix). A greater detail of the participant's responses on various survey scales regarding their demographic variables using the Kruskal-Wallis test and Mann-Whitney post-hoc multiple comparisons between groups is summarised in (S7 Appendix).

## Correlation between quality results scale and other accreditation scales

There was a significant positive correlation between healthcare professionals' perception of Quality Results (P < 0.001) and their perception of all other accreditation survey scales. The correlation between Customer or Patient Satisfaction and Quality Results was ranked the highest (γ = 0.611, P < 0.001), while the correlation between Awareness of the Accreditation and Quality Results was ranked the lowest (γ = 0.502, P < 0.001). Table 3 shows the correlation between Quality Results Scale and Other Accreditation Scales.

## Association between the dependent variable and independent variables

Overall, the simple linear regression analysis revealed a statistically significant positive relationship between Quality Results (dependent variable) and the accreditation survey's eight scales (independent variables) (Table 4). For instance, a simple linear regression was performed to investigate if Strategic Quality Planning significantly predicted Quality Results. The regression results showed that the model explained ($R^2$ = 0.305) of the variance and was significant (F = 162.654, P < 0.001). The slope coefficient for Strategic Quality Planning, holding for all other variables constant, was (β= 0.426, P < 0.001), so the Quality Results increased by (0.426) for each extra unit of Strategic Quality Planning. Likewise, the other independent variables were increased by a similar margin (Table 4).

A multiple regression (model 1) showed a significant relationship between Quality Results and Customer or Patient Satisfaction (P < 0.001), Management and Leadership (P = 0.030), and Accreditation Impact (P = 0.001) scales. Analysis of multiple regression (model 2) also showed a significant relationship between Quality Results and the same scales above; Customer or Patient Satisfaction (P < 0.001), Management and Leadership (P = 0.037), and

**Table 3. Correlation between quality results scale and other accreditation scales.**

| Scale | Quality Results | |
|---|---|---|
| | Correlation Coefficient | P-Value |
| Strategic Quality Planning | 0.527 | < 0.001* |
| Human Resource Utilization | 0.521 | < 0.001* |
| Quality Management | 0.546 | < 0.001* |
| Customer or Patient Satisfaction | 0.611 | < 0.001* |
| Management and Leadership | 0.516 | < 0.001* |
| Accreditation | | |
| Accreditation Impact | 0.586 | < 0.001* |
| Staff Involvement in Accreditation | 0.557 | < 0.001* |
| Awareness of the Accreditation | 0.502 | < 0.001* |

*. Indicates significant P-values.

Accreditation Impact (P < 0.001). These three scales were found to be the most important accreditation predictors of better-Quality Results. Each extra unit of the score on Customer or Patient Satisfaction, Management and Leadership, and Accreditation Impact increased the score on quality results by 0.263, 0.084, and 0.187, respectively. The $R^2$ value was 0.588, so approximately 58.8% of the variation in Quality Results can be explained by the model containing the independent variables. However, none of the demographic variables were found to be a contributing factor to these results or a predictor of better-Quality Results (Table 5).

## Discussion

To the best of our knowledge, this is the first study set out to investigate healthcare professionals' perceptions about implementing accreditation as a strategy to improve the quality of healthcare services and organisational performance in Jordan's public hospitals. The study findings showed that the majority of healthcare professionals agree that accreditation had improved the quality of healthcare services and hospitals' performance in a wide range of clinical and administrative aspects.

The study found that the mean scores of the respondents' evaluations of accreditation impact on the quality of healthcare services were almost high on all scales. This indicated that healthcare professionals have a positive perception and favourable view towards accreditation.

**Table 4. Results of simple linear regression model testing for the predictors of quality results.**

| Scale | R Square | F-value | β (Beta) (Std. error) | P-value |
|---|---|---|---|---|
| Strategic Quality Planning | 0.305 | 162.654 | 0.426 (0.033) | < 0.001* |
| Human Resource Utilization | 0.314 | 169.111 | 0.342 (0.026) | < 0.001* |
| Quality Management | 0.357 | 205.005 | 0.419 (0.029) | < 0.001* |
| Customer or Patient Satisfaction | 0.489 | 354.486 | 0.538 (0.029) | < 0.001* |
| Management and Leadership | 0.403 | 250.087 | 0.450 (0.028) | < 0.001* |
| Accreditation | | | | |
| Accreditation Impact | 0.454 | 307.894 | 0.533 (0.030) | < 0.001* |
| Staff Involvement in Accreditation | 0.359 | 207.063 | 0.454 (0.032) | < 0.001* |
| Awareness of the Accreditation | 0.274 | 139.955 | 0.381 (0.032) | < 0.001* |

*. Indicates significant P-values.

**Table 5. Results of multiple regression analysis for the predictors of quality results.**

| Scale | Model 1 | | Model 2 | |
|---|---|---|---|---|
| | Beta (Std. error) | P-value | Beta** (Std. error) | P-value |
| Strategic Quality Planning | 0.064 (0.039) | 0.099 | 0.069 (0.039) | 0.077 |
| Human Resource Utilization | 0.056 (0.032) | 0.087 | 0.056 (0.033) | 0.086 |
| Quality Management | 0.010 (0.040) | 0.814 | -0.003 (0.043) | 0.951 |
| Customer or Patient Satisfaction | 0.260 (0.044) | < 0.001* | 0.263 (0.046) | < 0.001* |
| Management and Leadership | 0.087 (0.040) | **0.030*** | 0.084 (0.040) | **0.037*** |
| Accreditation | | | | |
| Accreditation Impact | 0.191 (0.052) | < 0.001* | 0.187 (0.053) | **0.001*** |
| Staff Involvement in Accreditation | 0.056 (0.053) | 0.290 | 0.070 (0.054) | 0.194 |
| Awareness of the Accreditation | -0.066 (0.046) | 0.148 | -0.082 (0.048) | 0.087 |
| Demographic Variables | | | | |
| Gender | | | -0.046 (0.038) | 0.233 |
| Age | | | -0.010 (0.033) | 0.757 |
| Working experience | | | 0.030 (0.028) | 0.285 |
| Level of education | | | 0.032 (0.029) | 0.273 |
| Occupation categories | | | 0.012 (0.008) | 0.151 |
| Hospital size | | | -0.013 (0.033) | 0.693 |

*. Indicates significant P-values.

**. The model was accounting for demographic variables.

Notes: (Model 1) $R^2$ = 0.579, F = 62.487, P < 0.001, (Model 2) $R^2$ = 0.588, F = 36.460, P < 0.001.

However, there is considerable variation between administrative staff and clinical professionals' perceptions about the aims and objectives of using accreditation as a strategy for healthcare quality improvement.

The findings of the study indicated that there was a significant positive correlation between healthcare professionals' perception of Quality Results (P < 0.001) and their perception of all other accreditation survey scales. The correlation between Customer or Patient Satisfaction and Quality Results was ranked the highest (γ = 0.611, P < 0.001), while the correlation between Awareness of the Accreditation and Quality Results was ranked the lowest (γ = 0.502, P < 0.001). Overall, regression analysis revealed a statistically significant positive relationship between Quality Results (dependent variable) and the accreditation survey's eight scales (independent variables). A multiple regression (model 1) showed a significant relationship between Quality Results and Customer or Patient Satisfaction (P < 0.001), Management and Leadership (P = 0.030), and Accreditation Impact (P = 0.001) scales, and these scales were significant predictors of better-quality results in healthcare services.

A survey was conducted in four public hospitals in Jordan, with a high response rate of nearly 74.4%. Our findings show the job distribution of respondents varied significantly across the participating hospitals, with nurses being the largest proportion of respondents, followed by physicians and administrators/managers. This may have an impact on the study results and the relative scores provided by healthcare professionals, particularly as some studies have shown considerable disparities in opinion of accreditation, particularly among nurses versus physicians, with nurses being more supportive of accreditation than physicians. This could impact the study results and healthcare professionals' scores [18,28,40].

The study found that occupational categories significantly predict various scales, with accreditation having the strongest relationship with hospital directors, administrative and

management staff, and nurses, respectively. Hospital management teams are more in agreement with accreditation than clinical teams (e.g., physicians), as administrative staff prepare compliance reports, verify records, and respond to accreditation requests [15,40]. This could also be because the hospital management views accreditation as a management tool for standardising work processes and improving organisational structure rather than a strategy for improving clinical quality outcomes and patient safety. For example, the study found that hospitals have effective policies to improve care quality and encourage staff to document quality issues. However, the feedback system is less effective in assessing future patient needs and expectations, suggesting, at least in part, that work is primarily used for accreditation purposes. This finding aligns with a previous study suggesting management teams may shift their focus from identifying improvement opportunities to meeting standards and obtaining accreditation status [46]. Future research should investigate the difference in perception between administrative staff and clinical providers about the aims and objectives of using accreditation as a tool for healthcare improvement.

For accreditation impact, findings demonstrate the majority of healthcare professionals agree that accreditation positively impacts outcomes in various administrative and clinical aspects of their hospitals. Accreditation improves patient care, staff motivation, teamwork, and implements changes, aligning with previous research on its positive impact [15,24,47–49,61].

Regarding human resources scale, the study found that only 40.1% of respondents were rewarded and recognised for participating in quality improvement interventions. Previous research highlights the importance of rewards and recognition in enhancing employee satisfaction, performance, and sustainability of improvements [9,25]. Recognition and rewards, such as financial incentives, can also increase motivation, foster teamwork, and encourage innovation among staff [47,50,51], ultimately improving the quality results and efficiency of HCOs.

Contrary to some previous studies [52–54], our findings reveal that accreditation is linked to higher patient satisfaction, especially in addressing complaints and assessing current needs and expectations. However, two-thirds of respondents believe that hospital feedback systems are ineffective for predicting future patient needs or designing new services. Hospital management and leadership must carefully plan and address future patients' needs, preferences, and expectations and regard them as valuable sources for strategic planning in hospitals to achieve long-term success in quality improvement efforts [40,55].

Concerning awareness of accreditation, our findings show the majority of hospital staff are aware that accreditation process in underway, but around one-third are unaware of its aims and objectives. Additionally, over half of respondents were unaware of recommendations made to their hospital since the last survey. This raises questions about the effectiveness of hospital communication systems and the importance of accreditation information distribution for all staff and units. Involving a proportion of staff in the accreditation process could lead to inconsistent healthcare quality and affect hospitals' ability to sustain improvements and accreditation status over time, a finding that is consistent with previous similar research [38,51,56]. To ensure successful accreditation implementation, healthcare leaders and policymakers must improve hospital communication systems, share accreditation knowledge with all departments, and encourage all staff participation.

A unique finding of this study is that 47% of respondents reported that patients are unaware of a hospital's accreditation process. This indicating there is a significant gap in patient awareness and engagement with quality improvement activities. Patient involvement is crucial for successful implementation of healthcare quality improvement interventions [40], and the efficacy of such interventions relies on strong relationships, communication systems,

and knowledge shared by all stakeholders, including patients [17,25]. Hospital management and leadership should prioritize the dissemination of accreditation information and patient engagement as an integral part of healthcare services. However, the impact of accreditation from the perspective of patients is an under-researched area that requires further attention in future accreditation research.

Finally, the study found that hospital management and leadership support are vital for the successful implementation of accreditation. While senior executives began to improve care quality and services based on feedback from staff, customers, and accrediting bodies, they were not consistently involved in activities and did not allocate sufficient resources for quality improvement. Many studies have emphasised the importance of hospital management and leadership as a driving force for quality improvement initiatives, providing staff with adequate resources and time to facilitate successful accreditation implementation [24,28,40]. Strong relationships between hospital leaders and healthcare professionals, as well as regular communication, are essential for enabling change within healthcare facilities [14]. The application of new quality improvement interventions, such as accreditation, is often associated with psychological tensions in healthcare facilities. Therefore, the continued support, commitment, and interaction of hospital management and leadership are vital to overcome these difficulties [47,57]. These findings confirm previous research indicating the key role healthcare leadership plays in the effective normalisation of accreditation and quality improvement initiatives in healthcare settings [51,58,59].

## Strengths and limitations

This study is the first evaluation of healthcare professionals' perceptions and views on hospital accreditation and its impact on quality improvement in Jordan's public hospitals. It is unique in its comprehensive coverage of Jordan's three geographical regions and four governorates, using primary data, a large sample size of 500, and a relatively high response rate of 74.4%. The study's strengths include its comprehensive nature and representativeness.

Two limitations should be considered while evaluating our results. Firstly, it only focused on healthcare professionals working for Jordan's Ministry of Health (MOH), potentially resulting in selection bias. This means the findings cannot be generalised to other hospitals in different contexts (e.g., private hospitals). Despite this, the study offers valuable insights into the implementation of accreditation based on MOH healthcare professionals' perceptions. Secondly, the findings are based on healthcare professionals' perceptions without additional analysis of clinical quality measures. This is because data on quality measures is not easily accessible in all public hospitals. Future research should evaluate the impact of accreditation on clinical quality outcomes.

## Conclusion

This study presents valuable insights into the perceptions of healthcare professionals toward accreditation and its impact in public hospitals in Jordan. The majority receive accreditation positively, recognizing its potential to improve the quality of healthcare and organizational performance. Though, there are subtle variations between administrative and clinical professionals. The study recommends encouraging accreditation as a strategy to enhance healthcare service quality and organisational performance. The study findings show a significant correlation between Quality Results and other survey scales such as Customer or Patient Satisfaction, Management and Leadership, and Accreditation Impact. These findings emphasize the importance of a comprehensive approach toward accreditation that highlight both clinical and organizational elements.

The study provides useful insights into the implementation of accreditation programs by healthcare professionals. The findings have significant implications for policymakers, hospital managers, and researchers in the future successful preparation and implementation of hospital accreditation. It emphasized the importance of enhancing the efficiency of hospitals' communication systems, sharing knowledge about accreditation among different stakeholders, and encouraging a greater number of staff to take part actively in embedding and normalizing accreditation standards into routine practice to address barriers to improving quality in healthcare services. The study also suggests that regular and effective communication, strong relationships between hospital leaders and healthcare professionals, and patient engagement as an integral part of healthcare services are all essential to enabling change within healthcare facilities.

Future research is needed to explore the long-term impact of accreditation and investigate strategies to enhance its effectiveness. Also, exploring accreditation within different healthcare contexts, such as a private healthcare sector (private hospitals). Finally, additional exploration of the contextual factors that impact the perceptions of healthcare professionals on accreditation, particularly among different healthcare professional groups. This will help provide policy direction and improve understanding of accreditation's impact.

## Supporting information

**S1 Appendix. The main characteristics of the case study sites.**
(DOCX)

**S2 Appendix. SQUIRE guidelines for reporting system-level work (accreditation) to improve the quality, safety, and value of healthcare.**
(DOCX)

**S3 Appendix. Minimum sample size and number of questionnaires that needed for each case in the study.**
(DOCX)

**S4 Appendix. A copy of the questionnaire.**
(DOCX)

**S5 Appendix. Participant's responses to the accreditation survey's various scales and subscales.**
(DOCX)

**S6 Appendix. Result of significant and non-significant associations of each survey scale with demographic variables.**
(DOCX)

**S7 Appendix (Table 1–Table 9). Analysis of mean ranks for each scale of the survey with the demographic variables.**
(DOCX)

## Author contributions

**Conceptualization:** William J. Jackson.

**Data curation:** Mohammad J. Alhawajreh, Audrey S. Paterson, William J. Jackson.

**Formal analysis:** Mohammad J. Alhawajreh.

**Investigation:** Mohammad J. Alhawajreh.

**Methodology:** Mohammad J. Alhawajreh, Audrey S. Paterson, William J. Jackson.

**Project administration:** Mohammad J. Alhawajreh.

**Resources:** Mohammad J. Alhawajreh, Audrey S. Paterson, William J. Jackson.

**Validation:** William J. Jackson.

**Writing – original draft:** Mohammad J. Alhawajreh.

**Writing – review & editing:** Audrey S. Paterson, William J. Jackson.

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
