## [Decision Letter · Decision Letter 0]

22 Jan 2025

PONE-D-24-56923Healthcare professionals' perceptions about implementing accreditation as a strategy to improve healthcare quality and organisational performance: a cross-sectional survey studyPLOS ONE

Dear Dr. Alhawajreh,

Thank you for submitting your manuscript to PLOS ONE. After careful consideration, we feel that it has merit but does not fully meet PLOS ONE’s publication criteria as it currently stands. Therefore, we invite you to submit a revised version of the manuscript that addresses the points raised during the review process.

We look forward to receiving your revised manuscript.

Kind regards,

Maher Abdelraheim Titi

Academic Editor

PLOS ONE

Journal Requirements:

2. Please note that in order to use the direct billing option the corresponding author must be affiliated with the chosen institute. Please either amend your manuscript to change the affiliation or corresponding author, or email us at plosone@plos.org with a request to remove this option.

Reviewers' comments:

Reviewer's Responses to Questions

**Comments to the Author**

1. Is the manuscript technically sound, and do the data support the conclusions?

Reviewer #1: Yes

Reviewer #2: Partly

2. Has the statistical analysis been performed appropriately and rigorously? 

Reviewer #1: Yes

Reviewer #2: Yes

3. Have the authors made all data underlying the findings in their manuscript fully available?

Reviewer #1: Yes

Reviewer #2: Yes

4. Is the manuscript presented in an intelligible fashion and written in standard English?

Reviewer #1: Yes

Reviewer #2: Yes

5. Review Comments to the Author

Reviewer #1: Thank you for the opportunity to review this manuscript, which explores an important and timely topic in healthcare management. Below, I provide detailed feedback (as requested) on the manuscript, addressing its technical soundness, statistical analysis, data availability, and overall clarity.

Introduction:

The introduction provides a strong starting point, effectively highlighting the significance of accreditation in healthcare. However, the study could benefit from being more explicitly positioned within the global context. Including contrasting findings from high-income countries, such as the US, Canada, or Australia, would provide valuable comparisons and enhance the discussion. Furthermore, while the rationale for focusing on public hospitals in Jordan is clear, expanding on how these findings could inform accreditation practices in similar healthcare systems would be beneficial. It’s worth noting that Jordan faces unique challenges, such as resource constraints, workforce shortages, and its role as a regional hub for refugees. Acknowledging these factors would strengthen the relevance of the study to other resource-limited settings.

Background:

The background lays a solid foundation by linking global healthcare challenges to the need for quality improvement strategies like accreditation. However, this section could be further strengthened by addressing the mixed evidence surrounding accreditation’s effectiveness. While the manuscript mentions conflicting findings, incorporating specific examples of studies showing both positive and critical outcomes would add balance. Additionally, the limited research on accreditation in the Eastern Mediterranean Region is highlighted but not fully explored. Expanding on the barriers to research in this region, such as data accessibility and resource limitations, would provide valuable context. A smoother transition connecting global challenges, regional gaps, and the study’s specific objectives would further enhance the narrative.

Methods:

The methods section is clear and comprehensive, with a well-defined study design and robust data collection process. The use of a web-based questionnaire and the strong response rate (74.4% of 500 invited participants) are commendable. However, the manuscript does not provide details on whether the questionnaire was adapted to reflect Jordan’s cultural or healthcare context. Addressing this point would enhance transparency and show that the instrument was appropriately tailored for the study population. Additionally, while the statistical methods are appropriate, a brief explanation of why non-parametric tests like Kruskal-Wallis were chosen over parametric alternatives would be useful. For instance, clarifying whether assumptions of normality or homogeneity of variance were tested and found to be violated would strengthen the methodological justification.

Results:

The results are presented clearly, with effective use of tables and figures to summarise key findings. The strong positive correlation between healthcare professionals’ perception of Quality Results and other accreditation scales is well-supported by statistical analysis. Briefly highlighting how observed differences between clinical and administrative staff might impact accreditation outcomes would add depth to the findings.

Discussion:

The discussion provides a thoughtful interpretation of the findings and successfully links them to broader improvements in healthcare quality and organisational performance. While the discussion acknowledges the study’s strengths, it could benefit from a more critical evaluation of its limitations, particularly the low patient awareness (47%) of accreditation. This finding has important implications for the effectiveness of accreditation programs and deserves further exploration. Additionally, engaging more deeply with contrasting findings from international studies would help contextualise the results within the broader accreditation literature. Although variations in perceptions between professional groups are mentioned, a more detailed analysis of these differences could further enrich the discussion.

Conclusion:

The conclusion effectively summarises the study’s main findings and underscores the positive impact of accreditation. However, I suggest including more actionable recommendations for policymakers and practitioners. For example, strategies to enhance patient awareness and better align perceptions between clinical and administrative staff could be discussed. Moreover, it would be helpful to explicitly acknowledge the study’s limitations, such as its focus on public hospitals, which may limit the generalisability of the findings. Proposing future research directions, such as exploring accreditation in private hospitals or other resource-limited settings, would strengthen this section.

Overall Evaluation:

This manuscript provides valuable insights into accreditation in the Eastern Mediterranean Region, filling an important gap in the literature. While the study is technically sound and well-structured, the suggested revisions would enhance its clarity, contextualisation, and overall impact. I recommend minor revisions to address the outlined concerns and ensure the manuscript reaches its full potential.

Reviewer #2: The study addresses an important topic and includes promising elements, such as studying four hospitals of different sizes, providing an interesting comparative analysis opportunity. However, several areas need significant improvement to enhance your work's clarity, rigor, and impact. Below are my detailed suggestions:

(1) References: Incorporate more recent studies, particularly those published in the last 3–5 years, to ensure the discussion reflects the latest hospital management and accreditation advancements (e.g., Araujo, C. A., Siqueira, M. M., & Malik, A. M. (2020). Hospital accreditation impact on healthcare quality dimensions: a systematic review. International Journal for Quality in Health Care, 32(8), 531-544.)

(2) Forced-Response Bias: The questionnaire design does not include a "Don't know" or "Not applicable" option and requires participants to respond to every question. This approach may introduce response bias, as participants may choose arbitrary answers when they lack knowledge or opinion on a particular question.

(3) Case Selection: Including two large and two small hospitals is a strength, but the manuscript does not explain the rationale behind their selection. Please elaborate on the criteria for choosing these hospitals and how their characteristics align with the study's objectives. Also, as proposed by Yin, this strategy of selecting units with different characteristics is very rich and should be used to compare the results and further understand the results in various contexts. The authors should take advantage of this choice. Another issue is the triangulation of the findings, which is essential in case study research to ensure the validity of conclusions.

(4) Lack of Practical Implications: The findings would benefit from further interpretation regarding practical implications for hospital management or policy. How can these results inform decision-making or operational improvements?

(5) Include three paragraphs in the Conclusion section to clarify the research contributions to academics, managers, and society. Please, one paragraph for each dimension.

6. PLOS authors have the option to publish the peer review history of their article (what does this mean?). If published, this will include your full peer review and any attached files.

Reviewer #1: No

Reviewer #2: **Yes: **Claudia Araujo

---

## [Author Response · Author response to Decision Letter 0]

27 Jan 2025

Response to Reviewers

Title:

Healthcare professionals' perceptions about implementing accreditation as a strategy to improve healthcare quality and organisational performance: a cross-sectional survey study

Version: 1

Date: Jan 24, 2025

Author's response to reviews: see over

Subject: Response to reviewer comments for the manuscript “PONE-D-24-56923”

Healthcare professionals' perceptions about implementing accreditation as a strategy to improve healthcare quality and organisational performance: a cross-sectional survey study.”

Thank you for the opportunity to revise and resubmit the above manuscript to your esteemed journal. We thank the academic editor and reviewer(s) for their detailed and thorough comments and have addressed their concerns in the revised version of the manuscript. Kindly find below a point-by-point description of the changes made to the manuscript.

Academic Editor

Comment 1:

Response 1:

Thank you for highlighting these requirements, this has now been completed.

Responses to reviewers’ comments:

Reviewer #1:

Thank you for the opportunity to review this manuscript, which explores an important and timely topic in healthcare management. Below, I provide detailed feedback (as requested) on the manuscript, addressing its technical soundness, statistical analysis, data availability, and overall clarity.

1. Introduction:

The introduction provides a strong starting point, effectively highlighting the significance of accreditation in healthcare. However, the study could benefit from being more explicitly positioned within the global context. Including contrasting findings from high-income countries, such as the US, Canada, or Australia, would provide valuable comparisons and enhance the discussion. Furthermore, while the rationale for focusing on public hospitals in Jordan is clear, expanding on how these findings could inform accreditation practices in similar healthcare systems would be beneficial. It’s worth noting that Jordan faces unique challenges, such as resource constraints, workforce shortages, and its role as a regional hub for refugees. Acknowledging these factors would strengthen the relevance of the study to other resource-limited settings.

Response:

Thank you for your comment, we have added a new paragraph within the introduction to present a clear position of healthcare accreditation within the global context; see lines 109-121.

Background:

The background lays a solid foundation by linking global healthcare challenges to the need for quality improvement strategies like accreditation. However, this section could be further strengthened by addressing the mixed evidence surrounding accreditation’s effectiveness. While the manuscript mentions conflicting findings, incorporating specific examples of studies showing both positive and critical outcomes would add balance. Additionally, the limited research on accreditation in the Eastern Mediterranean Region is highlighted but not fully explored. Expanding on the barriers to research in this region, such as data accessibility and resource limitations, would provide valuable context. A smoother transition connecting global challenges, regional gaps, and the study’s specific objectives would further enhance the narrative.

Response:

As per your suggestion, we have added three paragraphs connecting global challenges, regional gaps, and the study’s specific objectives; see lines 148-170.

2. Methods:

The methods section is clear and comprehensive, with a well-defined study design and robust data collection process. The use of a web-based questionnaire and the strong response rate (74.4% of 500 invited participants) are commendable. However, the manuscript does not provide details on whether the questionnaire was adapted to reflect Jordan’s cultural or healthcare context. Addressing this point would enhance transparency and show that the instrument was appropriately tailored for the study population. Additionally, while the statistical methods are appropriate, a brief explanation of why non-parametric tests like Kruskal-Wallis were chosen over parametric alternatives would be useful. For instance, clarifying whether assumptions of normality or homogeneity of variance were tested and found to be violated would strengthen the methodological justification.

Response:

Thank you for these comments, there was no need for questionnaire adaptation point was added; see lines 247-251. The assumption of normality violation was clarified; see lines 300-302.

3. Results:

The results are presented clearly, with effective use of tables and figures to summarise key findings. The strong positive correlation between healthcare professionals’ perception of Quality Results and other accreditation scales is well-supported by statistical analysis. Briefly highlighting how observed differences between clinical and administrative staff might impact accreditation outcomes would add depth to the findings.

Response:

The section of breakouts of results of the survey scales by demographic variables has addressed the impact of occupational categories on various survey scales; see lines 395-404.

4. Discussion:

The discussion provides a thoughtful interpretation of the findings and successfully links them to broader improvements in healthcare quality and organisational performance. While the discussion acknowledges the study’s strengths, it could benefit from a more critical evaluation of its limitations, particularly the low patient awareness (47%) of accreditation. This finding has important implications for the effectiveness of accreditation programs and deserves further exploration. Additionally, engaging more deeply with contrasting findings from international studies would help contextualise the results within the broader accreditation literature. Although variations in perceptions between professional groups are mentioned, a more detailed analysis of these differences could further enrich the discussion.

Response:

- Regarding the comment of low patient awareness (47%) of accreditation, this has been addressed, see lines 530- 539. (A unique finding of this study is that….).

- Limitations of our study were mentioned clearly in a separate section under the title Strengths and limitations , see line 564-572 .

- In terms of delving deeper into contrasting findings from international studies, the discussion section adequately addressed the opposing findings from international literature, as many examples of comparing our results to the international literature were provided.

- Regarding the variations in perceptions between professional groups, these were highlighted frequently within the discussion section. For example, see 480- 495 (The study found that occupational categories…..), lines 510-516 (Contrary to some previous studies….), lines 541-555 (Finally, the study found that hospital management and leadership)

5. Conclusion:

The conclusion effectively summarises the study’s main findings and underscores the positive impact of accreditation. However, I suggest including more actionable recommendations for policymakers and practitioners. For example, strategies to enhance patient awareness and better align perceptions between clinical and administrative staff could be discussed. Moreover, it would be helpful to explicitly acknowledge the study’s limitations, such as its focus on public hospitals, which may limit the generalisability of the findings. Proposing future research directions, such as exploring accreditation in private hospitals or other resource-limited settings, would strengthen this section.

Response:

Thank you for this feedback,

- Regarding more actionable recommendations for policymakers and practitioners, see lines 586- 596 (The study provides valuable insights into the implementation….)

- Regarding the study’s limitations, it has been addressed explicitly, see lines 5564- 572 (Two limitations should be considered while evaluating our results…)

- As per your suggestion we have added another point regarding future research directions; see lines 598-603 (Also, exploring accreditation….)

Reviewer #2:

The study addresses an important topic and includes promising elements, such as studying four hospitals of different sizes, providing an interesting comparative analysis opportunity. However, several areas need significant improvement to enhance your work's clarity, rigor, and impact. Below are my detailed suggestions:

(1) References: Incorporate more recent studies, particularly those published in the last 3–5 years, to ensure the discussion reflects the latest hospital management and accreditation advancements (e.g., Araujo, C. A., Siqueira, M. M., & Malik, A. M. (2020). Hospital accreditation impact on healthcare quality dimensions: a systematic review. International Journal for Quality in Health Care, 32(8), 531-544.)

Response:

As per your suggestion, recent studies have been cited to reflect the up-to-date influence of healthcare accreditation od different aspects healthcare quality, such as (Araujo et al., 2020), see line 500 (implements changes, aligning with previous research…..), (Filip et al., 2022), and others.

(2) Forced-Response Bias: The questionnaire design does not include a "Don't know" or "Not applicable" option and requires participants to respond to every question. This approach may introduce response bias, as participants may choose arbitrary answers when they lack knowledge or opinion on a particular question.

Response:

Thank you for pointing out this important issue. We agree that the absence of 'Don't know' and 'Not applicable' options could introduce response bias. The initial rationale for excluding these options was to encourage respondents to consider their position. However, to mitigate the potential impact of this bias, we carefully analysed response patterns to identify potential inconsistencies and focused on overall trends rather than individual responses in different survey scales.

(3) Case Selection: Including two large and two small hospitals is a strength, but the manuscript does not explain the rationale behind their selection. Please elaborate on the criteria for choosing these hospitals and how their characteristics align with the study's objectives. Also, as proposed by Yin, this strategy of selecting units with different characteristics is very rich and should be used to compare the results and further understand the results in various contexts. The authors should take advantage of this choice. Another issue is the triangulation of the findings, which is essential in case study research to ensure the validity of conclusions.

Response:

- Regarding the criteria for choosing participating hospitals and how their characteristics align with the study's objectives, more elaboration about criteria of selecting hospitals was presented in study design and setting section, see lines 194-219.

- We discussed and compared the key findings of the study with the international literature to provide in-depth understanding of the accreditation impact on the quality of healthcare.

(4) Lack of Practical Implications: The findings would benefit from further interpretation regarding practical implications for hospital management or policy. How can these results inform decision-making or operational improvements?

Response:

Thank you for this comment. Practical implications for hospital management or policy were mentioned in the conclusion section, see lines 586-596.

(5) Include three paragraphs in the Conclusion section to clarify the research contributions to academics, managers, and society. Please, one paragraph for each dimension.

Response:

Thank you for this valuable feedback, we rewrote the conclusion section to address the study implications for policymakers, hospital managers, and researchers, see lines 586-603.

Thank you for providing us with the opportunity to address the reviewers’ comments. We look forward to the outcome of the review process.

Best regards,

Authors

---

## [Decision Letter · Decision Letter 1]

24 Feb 2025

Healthcare professionals' perceptions about implementing accreditation as a strategy to improve healthcare quality and organisational performance: a cross-sectional survey study

PONE-D-24-56923R1

Dear Dr. Alhawajreh,

We’re pleased to inform you that your manuscript has been judged scientifically suitable for publication and will be formally accepted for publication once it meets all outstanding technical requirements.

Kind regards,

Maher Abdelraheim Titi

Academic Editor

PLOS ONE

Additional Editor Comments (optional):

Reviewers' comments:

Reviewer's Responses to Questions

**Comments to the Author**

1. If the authors have adequately addressed your comments raised in a previous round of review and you feel that this manuscript is now acceptable for publication, you may indicate that here to bypass the “Comments to the Author” section, enter your conflict of interest statement in the “Confidential to Editor” section, and submit your "Accept" recommendation.

Reviewer #1: All comments have been addressed

Reviewer #2: All comments have been addressed

2. Is the manuscript technically sound, and do the data support the conclusions?

Reviewer #1: Yes

Reviewer #2: Yes

3. Has the statistical analysis been performed appropriately and rigorously? 

Reviewer #1: Yes

Reviewer #2: Yes

4. Have the authors made all data underlying the findings in their manuscript fully available?

Reviewer #1: Yes

Reviewer #2: Yes

5. Is the manuscript presented in an intelligible fashion and written in standard English?

Reviewer #1: Yes

Reviewer #2: Yes

6. Review Comments to the Author

Reviewer #1: The authors have addressed all the review comments thoughtfully and comprehensively. The revisions improve the clarity, contextualisation, and methodological transparency of the manuscript. Key additions, such as the expanded discussion on global accreditation practices, the clarification of methodological choices, and the enhanced recommendations for policymakers, have strengthened the study’s contribution.I have no further comments.

Reviewer #2: Thanks for the reviewed version. The authors have addressed all the comments and made the necessary changes in the manuscript.

7. PLOS authors have the option to publish the peer review history of their article (what does this mean?). If published, this will include your full peer review and any attached files.

Reviewer #1: No

Reviewer #2: **Yes: **Claudia Araujo

---

## [Editor Report · Acceptance letter]

PONE-D-24-56923R1

PLOS ONE

Dear Dr. Alhawajreh,

I'm pleased to inform you that your manuscript has been deemed suitable for publication in PLOS ONE. Congratulations! Your manuscript is now being handed over to our production team.

Kind regards,

on behalf of

Dr. Maher Abdelraheim Titi

Academic Editor

PLOS ONE